# Barriers to Small Molecule Drug Discovery for Systemic Amyloidosis

**DOI:** 10.3390/molecules26123571

**Published:** 2021-06-11

**Authors:** Gareth J. Morgan

**Affiliations:** Section of Hematology and Medical Oncology, Amyloidosis Center, Department of Medicine, School of Medicine, Boston University, Boston, MA 02118, USA; gjmorgan@bu.edu

**Keywords:** systemic amyloidosis, amyloid fibrils, amyloidogenesis inhibitors, antibody light chains, light-chain stabilizers, doxycycline, EGCG, thioflavin T, filter trap, PAINS

## Abstract

Inhibition of amyloid fibril formation could benefit patients with systemic amyloidosis. In this group of diseases, deposition of amyloid fibrils derived from normally soluble proteins leads to progressive tissue damage and organ failure. Amyloid formation is a complex process, where several individual steps could be targeted. Several small molecules have been proposed as inhibitors of amyloid formation. However, the exact mechanism of action for a molecule is often not known, which impedes medicinal chemistry efforts to develop more potent molecules. Furthermore, commonly used assays are prone to artifacts that must be controlled for. Here, potential mechanisms by which small molecules could inhibit aggregation of immunoglobulin light-chain dimers, the precursor proteins for amyloid light-chain (AL) amyloidosis, are studied in assays that recapitulate different aspects of amyloidogenesis in vitro. One molecule reduced unfolding-coupled proteolysis of light chains, but no molecules inhibited aggregation of light chains or disrupted pre-formed amyloid fibrils. This work demonstrates the challenges associated with drug development for amyloidosis, but also highlights the potential to combine therapies that target different aspects of amyloidosis.

## 1. Introduction

Deposition of amyloid fibrils derived from antibody light-chain (LC) proteins is associated with organ damage in the disease amyloid light-chain (AL) amyloidosis [1,2]. AL amyloidosis is the most commonly diagnosed form of “systemic” amyloidosis, where amyloid fibrils form in multiple tissues [3]. In these diseases, amyloid fibrils are closely associated with pathology, and inhibition of amyloid deposition by suppression or stabilization of the precursor protein can lead to clinical benefits [4]. Here, “amyloidosis” refers to the disease and “amyloidogenesis” refers to the biochemical process of amyloid formation from soluble precursor proteins. “AL” is often used as a shorthand for AL amyloidosis, but strictly refers to the amyloid protein in its fibrillar state [3].

In AL amyloidosis, the precursor LCs are secreted from a clonal population of B lymphocytes, most often plasma cells in the bone marrow [1]. Amyloid fibrils are formed from a single “monoclonal” LC with a sequence that is unique to each patient. LCs are ~215-residue proteins that form two structural immunoglobulin domains. The N-terminal variable (V_L_) domain, which forms the structured core of AL amyloid fibrils, is the location of most of the diversity between LCs. The C-terminal constant (C_L_) domain is much more conserved and has a scaffolding and chaperone function. LCs can form homodimers, which may be stabilized by an inter-chain disulfide bond between the C_L_-domains.

Preventing amyloid deposition in AL amyloidosis is a longstanding goal of research and treatment. Cytotoxic chemotherapy can kill the clonal plasma cells, thus reducing levels of monoclonal LCs below that necessary for aggregation [5]. However, many patients are diagnosed after the onset of organ failure, which complicates treatment [6]. Therefore, substantial efforts are devoted to developing therapeutic strategies that could benefit these patients [7].

Amyloidogenesis by folded proteins is a complex, multi-step process (Figure 1), and pharmacologic intervention at any of these points could potentially benefit patients. Importantly, these steps could be targeted in combination, which could potentially lead to synergy between approaches. Several small molecules have been suggested as potential therapies for different amyloid diseases. Given the high cost of drug development, most research has focused on natural products or existing drugs. These molecules potentially offer a starting point for medicinal chemistry approaches to develop more efficacious molecules. Despite promising reports of efficacy in in vitro assays of amyloidogenesis, or even in clinical trials, intense lead optimization has not generally been carried out. One reason for this lack of optimization is the complexity of amyloidogenesis. It has been difficult to determine exactly which step or steps are affected by a particular molecule, and their precise molecular targets are not known.

Other than eradication of the producer cells, there are several potential mechanisms by which a drug could benefit individuals with AL amyloidosis. One strategy is to suppress unfolding of the protein and thereby reduce the levels of amyloid-competent species. This approach has been clinically efficacious in transthyretin amyloidosis, where the “kinetic stabilizer” drug tafamidis is approved in multiple regions [8,9]. Several molecules have been proposed as stabilizers of LCs [10,11,12] and we are actively pursuing this strategy. Alternatively, a drug could interfere with the self-association of non-native amyloid precursors to suppress self-association and amyloid formation. This approach has been explicitly pursued for several proteins, including α-synuclein [13] and insulin [14], and is proposed as a mechanism of action of other molecules. A similar approach is the capping of pre-existing amyloid fibrils to prevent their extension [15]. A third group of potential therapies aims to dissociate preexisting amyloid fibrils, either via small molecules, antibodies, or a combination of the two. Monoclonal antibodies directed at either the fibrils themselves [16,17] or the accessory protein serum amyloid P component (SAP) [18,19] have been used in clinical trials to recruit phagocytic cells to clear the amyloid. These strategies are outlined in Table 1.

In a typical drug development effort, high-throughput screening is used to identify molecules with activity in one or more assays, and counter-screening is used to remove the inevitable false-positive hit molecules. Counter-screening is critically important when introducing new molecules to an assay, because of the possibility for unexpected interactions between the small molecule and the system. “Pan-assay interference compounds” (PAINS) have been identified as recurrently problematic in multiple assays [20], and many molecules can also interfere with assays by forming colloidal aggregates [21]. Once a molecule has been identified as a true-positive hit, chemical analogs are synthesized with the aim of improving the efficacy of the molecule in assays, or optimizing other desirable properties. Anti-amyloidogenic activity in vitro is likely a necessary property for a potential drug for amyloidosis, but this activity must also be maintained in vivo. A candidate drug must also be sufficiently bioavailable and non-toxic at the concentrations required for activity in humans. Each of these properties can potentially be optimized, but this optimization is challenging without detailed information about a molecule’s target and mechanism of action.

The action of drug-like small molecules on amyloidogenesis is often assessed by their ability to prevent amyloid formation in vitro. However, molecules could in principle act anywhere along the reaction pathway depicted in Figure 1. More problematically, experimental artifacts could occur at any point along this pathway. Observations that certain molecules, such as doxycycline, appear to inhibit the aggregation of multiple proteins [22,23,24] may imply the existence of common protein structures that represent a general molecular target for amyloidosis. Similarly, reports that diverse small molecules can inhibit amyloidogenesis of proteins with immunoglobulin folds (including LCs and β_2_-microglobulin) may indicate that for each protein, there are multiple potential target structures [10,12,23,25]. However, these observations could also represent recurring experimental artifacts in multiple systems.

This work aims to identify the mechanisms by which small molecules could inhibit amyloidogenesis of antibody LCs. We have previously used high-throughput screening to identify small molecules that stabilize native LCs [10]. Here, a “drug repurposing” approach was investigated, using molecules that have been extensively studied in other amyloid systems. Eight small molecules, which have been suggested to inhibit some aspect of amyloidogenesis in either LCs or other amyloid precursor proteins, were tested in a series of simple assays designed to probe different potential mechanisms of action. The goals were to ask whether previous results could be recapitulated in experimental systems established in our laboratory, and if so, to determine the relevant mechanism of action. Identifying such a mechanism of action could allow a small molecule’s effects to be optimized via medicinal chemistry and identify molecules with complementary activities. A further goal was to test whether these molecules have a sufficiently large, true-positive effect that they would be selected and validated in a high-throughput screen.

## 2. Results

In order for LCs to form amyloid fibrils, productive interactions must occur between rare, transiently populated LC structural species. The steady-state concentration of these species is defined by both the rate of LC secretion from plasma cells and the stability of the LC, which is its propensity to remain in its folded state. Post-translational modifications may alter these processes. Most clearly, proteolytic cleavage of LCs can lead to fragments which are more amyloidogenic [26,27]. Full-length LCs are more stable in vitro and less prone to aggregation under native-like conditions than their component V_L_-domains [26,28]. AL fibrils appear to be primarily composed of fragments of LCs, although the nature of this cleavage is unknown and may happen after deposition as amyloid [27,29,30,31,32,33]. The nascent amyloid fibrils must be stable enough to recruit new protein molecules and grow into mature fibrils, rather than dissociate. These fibrils deposit in tissues and form interactions with other molecules, such as components of the extracellular matrix and SAP, which further stabilizes them. Finally, fibrils must be resistant to clearance by cells including macrophages that would normally remove macroscopic debris. The fine details of how these processes play out in an individual patient likely depend on the unique sequence of the monoclonal LC.

Other than eradication of the producer cells, there are several potential mechanisms by which a drug might inhibit amyloidogenesis. Since self-assembly of LCs requires multiple steps which cannot be easily resolved, the approach here was to look for large effects on amyloidogenesis using simple assays, which could then potentially be resolved further. LCs derived from the *IGLV6-57* gene, which is over-represented in AL amyloidosis, were studied. Three mechanisms are investigated: stabilization of the full-length LC native state, inhibition of V_L_-domain self-assembly, and dissolution of pre-formed V_L_-domain amyloid fibrils. The differences between these mechanisms can be visualized using a simplified free energy diagram (Figure 2), which makes two key assumptions. Amyloid is assumed to be the most stable state at equilibrium, and only a single transition state, corresponding to self-assembly of non-native species, is assumed to be rate-limiting. This transition state model is an approximation of the multiple transition states that would be necessary to fully describe amyloidogenesis. The equilibrium stability of amyloid relative to soluble LCs is defined as ΔG_fibril_, which depends on concentration. The limit of solubility, or critical concentration, is defined as the total concentration of LC where ΔG_fibril_ is zero. Each potential inhibitory mechanism represents the interaction of a small molecule with a single LC conformational state. This interaction alters the relative free energy of that state, leading to distinct consequences. Stabilization of the native state would lead to a decrease in the rate of aggregation and a decrease in the equilibrium concentration of amyloid. Inhibition of self-association would also decrease the rate of aggregation, but would not alter the equilibrium concentration of amyloid. Dissolution or destabilization of amyloid fibrils would reduce the equilibrium concentration of amyloid but not alter the rate at which it is formed. These energetic properties mean that it should be possible to distinguish how small molecules alter amyloidogenesis using a kinetic assay that detects amyloid fibrils, such as measuring the fluorescence of thioflavin T (ThT), a benzothiazole dye with environment-sensitive fluorescence properties whose fluorescence increases when bound to amyloid fibrils. However, if amyloid is substantially more stable than native LCs, even a large free energy change in the presence of a small molecule may not lead to a measurable change in the amount of amyloid present at equilibrium. Measuring the amounts of soluble and insoluble LC, both at the endpoints of the aggregation reaction and following incubation of pre-formed fibrils with small molecules, allows these mechanisms to be distinguished more readily. Deviations from this model would indicate more complex mechanisms of aggregation or of small molecule action.

Eight molecules were studied (Figure 3): (1) Coumarin 1, which was identified as a stabilizer of native LCs [10]. (2) Doxycycline, a tetracycline antibiotic, which has been proposed to inhibit amyloid formation by several proteins, including LCs, and disrupt amyloid fibrils [22,23,24,34]. (3) Oxidized and (4) reduced forms of epigallocatechin gallate (EGCG), which have been proposed to remodel amyloid fibrils [35,36] and have been used in clinical trials for AL amyloidosis [37]. EGCGox and EGCGred refer to pre-oxidized and freshly thawed EGCG, respectively. (5) Rifampicin and (6) rifamycin SV, two related antibiotics, which have been shown in vitro to interfere with self-association of β_2_-microglobulin, an amyloid-forming protein with a C_L_-domain-like secondary structure [25]. (7) Methylene blue, which has been shown to inhibit formation of amyloid fibrils by tau [38] and LCs [12]. (8) Diflunisal, a stabilizer of the native state of tranthyretin [39,40] is not expected to interact with LCs and was included as a negative control. LCs derived from the IGLV6-57 gene, which is over-represented in AL amyloidosis [41], are used in all experiments. For each experiment, a different LC was used in order to maximize the signal that was measured, as described below. In all experiments, LC concentration was 5 µM (monomer equivalent; 0.06 mg/mL V_L_-domain or 0.12 mg/mL full-length LC) and small molecule concentration was 50 µM. All experiments were carried out in phosphate-buffered saline (PBS) at pH 7.4 and 37 °C. Doxycycline, EGCG, and methylene blue are soluble in water at the stock concentration of 5 mM, and other molecules were dissolved in ethanol. To verify that the ethanol had no influence on the LCs, additional controls using 1% (*v*/*v*) ethanol vehicle were used in each experiment.

### 2.1. Native State Stabilization

Amyloid fibrils are non-native assemblies, so unfolding from the LC native state, during or after secretion, is required for amyloid formation. The rate at which LCs unfold to form non-native species is described by their kinetic stability, which describes the energy barrier to unfolding [26]. Limited proteolysis [42] is a method to measure kinetic stability in a disease-relevant manner, since proteolysis appears to be important for AL amyloidosis pathogenesis. Non-native LC conformations are more susceptible to endoproteolysis than native LCs because endoproteases preferentially bind to unstructured peptide substrates. Since proteolysis is irreversible and faster than refolding, the rate at which a folded protein is degraded is determined by its unfolding rate, a situation known as EX1 kinetics (Figure 4a). Stabilization was assessed using an unstable full-length LC derived from a patient with AL, known as WIL [43], which we previously showed could be stabilized by small molecules [10]. Using this unstable LC maximizes the potential signal of the experiment, which is defined by the extent to which unliganded LCs are degraded over the incubation period. This approach of measuring at a single timepoint to infer the kinetics of a well-defined system is a way to minimize the number of measurements required for an experiment, which is important for high-throughput screening.

LCs were incubated with proteinase K, a highly active protease that cleaves after aliphatic residues. Using this protease allows for detection of transient unfolding of regions of LCs that may lack cleavage sites for a less promiscuous protease. Three replicates were measured. After 4 h of incubation at 37 °C, reactions were quenched with a covalent protease inhibitor and residual full-length LCs were measured by SDS-PAGE. The extent of degradation of LCs in the presence or absence of small molecules is shown in Figure 4b. Of the molecules tested, only coumarin 1 consistently protected the LC against proteolysis. This observation is in agreement with the results of a previous high-throughput screen, where we verified that coumarin 1 binds directly to LCs and does not inhibit proteinase K [10]. The degradation of LCs in the presence of other small molecules supports the hypothesis that these molecules do not act by binding to the LC native state.

### 2.2. Inhibition of Amyloidogenesis

A potential drug for amyloidosis could specifically inhibit self-association of amyloid-competent species, affecting early initiation steps, subsequent elongation of fibrils, or both processes (*k1_on_*, *k1′_on_*, or *k2_on_* in Figure 1). Successful inhibition of any of these processes would lead to a reduction in the rate of amyloidogenesis (Figure 2), which could then be followed up with further experiments. For LCs, this self-association occurs between non-native structural conformations that either exist in equilibrium with the native state or can be formed following proteolysis. Several potential mechanisms by which such inhibition could occur are described in Figure 5. These mechanisms all involve binding of small molecules to transiently populated LC species, increasing the energy barrier to amyloidogenesis. Since these mechanisms would not alter the free energy difference between the native and fibrillar LC (Figure 2), there would be no reduction in the endpoint fraction of amyloid fibrils. However, the rate at which amyloid is formed would be reduced, possibly to the extent that no fibrils are formed over the course of an experiment.

Rational design of molecules that inhibit protein self-association is difficult because the properties of the protein species that self-assemble are not known, and these species are only transiently present in solution, often at very low concentration. Furthermore, designing inhibitors of specific protein–protein interfaces is challenging because these interfaces often comprise multiple weak, non-specific interactions. Efforts to identify inhibitors of amyloidogenesis therefore often rely on screening libraries of compounds with assays that detect amyloid fibril formation, such as fluorescence of ThT. Molecules that cause a reduction in ThT fluorescence after incubation with aggregation proteins would be considered hits. However, these assays are prone to false positives because displacement of ThT from fibrils or quenching of fluorescence could be interpreted as inhibition of amyloid formation (Figure 5c). Furthermore, the emission of fibril-bound ThT is not well-understood and can differ between batches of fibrils. Aggregation often proceeds with “sigmoidal” kinetics, where a “lag” phase is followed by a rapid increase in ThT fluorescence, which is a result of a reduced energy barrier to fibril extension (Figure 2b). Molecules that extend the lag phase may be more likely to be true-positive hits than those that reduce the final fluorescence intensity of the aggregate-bound ThT.

A number of molecules have been proposed to alter the morphology of the aggregates formed in vitro. However, whether these structural transformations are accessible to amyloid fibrils in patients is not known. If amyloid fibrils are highly stable under physiological conditions and LC concentrations, interconversion between alternate aggregate conformations is likely to be slow. This can be visualized by large energy barriers between different aggregate conformations (Figure 5a), corresponding to discrete fibril conformations that are only accessible via soluble, unfolded LCs. Non-fibrillar assemblies, which are often called “amorphous aggregates”, may be less stable and more accessible at equilibrium, and some small molecules appear to be able to direct protein aggregation towards such assemblies. Moreover, the mechanisms by which amyloid fibrils cause tissue damage are poorly understood, so it is not clear whether species such as amorphous aggregates, “spherical oligomers,” or “protofibrils” actually represent a less toxic endpoint than amyloid. These structures may be an additional source of variability in ThT experiments, since ThT may bind to different structures with different affinity. Due to these potential artifacts, the results of a ThT-binding experiment should therefore be verified by measuring a property such as residual soluble protein.

To ask whether small molecules can inhibit amyloidogenesis, their effect on LC V_L_-domain aggregation was measured in vitro by ThT kinetics (Figure 6). The LC V_L_-domain used is known as JTO-V_L_, which was originally identified in an individual with multiple myeloma, and which reproducibly forms amyloid fibrils in vitro [26,43]. Full-length LCs derived from the *IGLV6-57* gene, including JTO, do not readily aggregate at neutral pH, but their isolated V_L_-domains do so [26]. JTO-V_L_ is not derived from a patient with amyloidosis, but it aggregates readily and is predominantly folded under the conditions used for aggregation [26,44]. JTO-V_L_ reproducibly aggregates with kinetics that can be fitted to mechanistic models, but the kinetics are sensitive to reaction conditions and there is some variability in the ThT fluorescence measurements, particularly in the endpoint fluorescence intensity [26]. To minimize the effects of such variability, aggregation reactions were carried out in parallel on the same microwell plate. Although microplate-based methods yield less well-resolved data than cuvette-based measurements, they allow side-by-side comparison of multiple small molecules, using replicates to quantify variability. Full-length JTO was used as a negative control, because this LC does not form ThT-binding aggregates under similar conditions [26,45].

In the absence of small molecules, JTO-V_L_ reproducibly forms ThT-binding aggregates after 8–12 h of incubation with continuous shaking (Figure 6). Five molecules—oxidized and reduced EGCG, rifampicin, rifamycin SV, and methylene blue—all suppress the rise in ThT fluorescence associated with amyloiodgenesis. Coumarin 1 increases the fluorescence signal before the initiation of aggregation, due to its own fluorescence. However, closer examination of the kinetic traces (Figure 6, right panels) reveals an increase in ThT fluorescence at a similar time to that observed in the absence of small molecules. These results appear to be consistent with the hypothesis that the small molecules destabilize or remodel amyloid fibrils, leading to less ThT-binding material at the end of the reaction (Figure 2).

To test whether these molecules disrupt the binding or fluorescence of ThT, additional samples of JTO-V_L_ in adjacent wells were allowed to aggregate for 24 h (grey curves in Figure 6), after which small molecules were added. The change in ThT fluorescence upon addition of rifamycin SV, rifampicin, and methylene blue is consistent with suppression of ThT fluorescence, rather than inhibition of aggregation (Figure 7).

At the endpoint of the aggregation reactions, the presence of soluble and insoluble LC aggregates was assessed by a “filter trap” differential membrane binding assay [46] (Figure 8). Samples are drawn through a cellulose acetate membrane, which traps large aggregates, then a cellulose nitrate membrane, which binds soluble protein. Protein binding to each membrane is measured by staining with amido black dye [47]. The ratio between the staining intensity of the two membranes can report on the relative solubility of the LC. Molecules which suppress aggregation would be expected to reduce the amount of insoluble protein and retain soluble protein. Following aggregation, staining of soluble LCs decreased substantially compared to the non-aggregated controls, while staining of insoluble LCs increased, consistent with the presence of aggregates. Several molecules, particularly methylene blue, are colored, and cause visible staining on one or both membranes, making quantitation less accurate. Despite this limitation, aggregated LC species are present in all samples (Figure 8b). There is little apparent loss of cellulose acetate membrane staining that would correspond to insoluble material in the presence of small molecules. Similarly, there is no increase in cellulose nitrate membrane staining indicative of increased levels of soluble LC. Notably, similar levels of aggregate are detected in both the wells where small molecules were added before aggregation (Figure 8c, orange symbols), and those where small molecules were added to pre-formed fibrils (Figure 8c, grey symbols). LC aggregates which formed in the presence of EGCG stained more intensely with amido black than other aggregates, which may indicate differences in aggregate morphology, consistent with the weak binding of these aggregates to ThT.

These data indicate that none of the small molecules tested prevent LC V_L_-domain aggregation under these conditions. It is possible that non-amyloid aggregates are formed, which do not bind to ThT, but soluble protein is not retained in solution.

### 2.3. Dissolution of Pre-Formed Amyloid Fibrils

A small molecule that destabilizes amyloid fibrils and enhances their dissolution could help to clear fibril deposits from patients. This potential mechanism of action is distinct from inhibition of LC self-association, since the molecular target of the small molecule is the fibrils rather than pre-fibrillar intermediates. However, a sufficiently large effect could be observed in a ThT kinetic experiment as a reduction in the endpoint ThT fluorescence (Figure 2). Although small molecules including EGCG and doxycycline have been proposed to “remodel” existing fibrils, the mechanism by which this could occur is not clear. Above the critical solubility concentration, amyloid fibrils are more stable than their soluble precursors (Figure 2). Molecules that preferentially bind to amyloid fibrils over native LCs would enhance the stability of the fibrils, the opposite of the intended effect. Any destabilization of amyloid could occur via interactions with unfolded species that are present at equilibrium. Protein subunits continually exchange between the fibrils and solution, so a molecule that stabilizes non-fibrillar conformations could lead to dissociation of amyloid, potentially favoring the formation of non-amyloid, “amorphous” aggregates. However, such interactions would also necessarily occur during amyloidogenesis, diverting protein molecules away from fibril-competent conformations and altering the observed reaction kinetics. Therefore, a small molecule that destabilizes fibrils would likely affect the results of a ThT kinetics experiment.

Although the mechanisms involved have not been characterized, dissociation or alteration of pre-formed amyloid fibrils has been observed in the presence of small molecules [22,24]. Therefore, the effect of small molecules on the solubility of LC V_L_-domain amyloid fibrils, which were formed in vitro before addition of small molecules, was tested. LC V_L_-domain amyloid fibrils from a relatively stable V_L_-domain, the “germline” *IGLV6-57* sequence, referred to as 6aJL2-V_L_ [48], were used to maximize the likelihood that any LC that dissociated from the fibril would remain folded in solution. Amyloid dissolution is likely to occur slowly because of the large free energy barrier to dissociation (Figure 2). Doxycycline has been reported to dissociate amyloid fibrils over a 15-day time period [22]. Therefore, duplicate fibril samples were incubated with small molecules for 7 and 15 days.

After 7 and 15 days of incubation at 37 °C, LC solubility was assessed by the filter trap assay (Figure 9). To maximize the detection of soluble or insoluble material, LCs were precipitated by centrifugation, and the supernatant and resuspended precipitate were analyzed separately for each time point (orange and blue symbols in Figure 9, respectively). A single sample of soluble LC was used as a positive control. Staining of membranes by small molecules impeded quantitation of solubility, as described above. Quantitation of total aggregate staining shows that samples incubated with small molecules do not have reduced levels of insoluble LC (Figure 9c). The proportion of aggregated LC can be inferred from the ratio of cellulose acetate staining to cellulose nitrate staining (Figure 9d), which shows no difference between samples incubated with or without any small molecule except for methylene blue. However, methylene blue preferentially stains cellulose nitrate (Figure 8), so this observation does not necessarily demonstrate dissociation of aggregates. Furthermore, apparent levels of aggregated LC were higher in the precipitate fraction of all samples, consistent with retention of aggregated LC. Overall, no sample showed evidence of reduced aggregate content following incubation.

## 3. Discussion

Inhibition of amyloid fibril formation is challenging, and it is perhaps surprising that so many small molecules have been reported to inhibit the amyloidogenesis of several proteins in vitro or have shown efficacy in clinical trials. Such observations with natural products or drugs approved for other indications raise the possibility that more efficacious inhibitors of amyloidogenesis could be developed. As a first step towards this goal, this study attempted to identify molecules that could inhibit amyloidogenesis of LCs, and the mechanisms by which these molecules did so. Table 2 summarizes the results of these experiments. In these model systems, no molecule led to suppression of LC aggregation, although five small molecules suppressed ThT fluorescence. These molecules—oxidized and reduced EGCG, methylene blue, rifampicin, and rifamycin SV—reduced the amplitude of the endpoint ThT fluorescence. In a typical high-throughput assay where only endpoint fluorescence was measured, these molecules would all have registered as hits. However, none of these molecules suppressed total aggregation, and three, methylene blue, rifampicin, and rifamycin SV, appeared to alter the fluorescence of ThT, possibly by displacing or quenching ThT fluorescence. These results highlight the importance of verifying the effects of small molecules on ThT aggregation experiments, using an orthogonal assay.

Since AL amyloid is derived from a LC protein that is normally folded, stabilization of the LC native state presents an additional opportunity for intervention, analogous to that used successfully for transthyretin amyloidosis [8,9]. Coumarin 1, which was previously identified as a stabilizer of LCs [10], was able to protect a full-length, unstable LC from unfolding and subsequent proteolysis (Figure 3). If the hypothesis that unfolding of LCs from their native state is required for aggregation is correct, molecules that stabilize LCs in a similar way to coumarin 1 could reduce aggregation and benefit patients. Importantly, although coumarin 1 is active in this assay, it has a relatively weak affinity for LCs, around 3 µM for full-length WIL, and is unlikely to be sufficiently potent to be active in vivo. Therefore, the weaker protection afforded by doxycycline (Figure 3) likely does not explain the activity of doxycycline in patients [49,50]. The details of how coumarin 1 and other hits from the screen bind to LCs have been identified [10,11], and this information has been used to design and synthesize new molecules which are more efficacious stabilizers of LCs [51].

The ability of small molecules to suppress amyloidogenesis or disrupt amyloid fibrils in vitro has supported ongoing clinical trials of doxycycline and EGCG for individuals with systemic amyloidosis. However, both molecules are pleiotropic and are known to affect multiple processes in vitro and in vivo. Clearer mechanistic data on how these molecules work could help to optimize their eventual clinical use.

Doxycycline is a widely prescribed tetracycline antibiotic, which has been shown to prevent amyloidogenesis and disrupt amyloid fibrils formed by several proteins in vitro [22,23,24]. Doxycycline reduced LC-associated toxicity in a nematode model [52]. Doxycycline also suppressed amyloid deposition in a mouse model of AL amyloidosis and inhibited amyloid formation, but not aggregation, of a full-length LC derived from the *IGKV1-33* gene [24]. The experimental conditions used to induce aggregation of this full-length LC included cycles of 4 h incubations at 65 °C [24], which may lead to different aggregate structures than the conditions used in the experiments described in Figure 6, Figure 7 and Figure 8. Doxycycline was also observed to alter the morphology of fibrils extracted from an individual with AL amyloidosis [24]. These observations are consistent with data presented in Figure 8 and Figure 9 because alternative aggregate species would still be detected in the filter trap assays.

Clinical studies have shown a beneficial effect of using doxycycline over other antibiotics in patients with AL amyloidosis [49,50,53]. However, the effects of doxycycline on amyloidogenesis in vitro have been most thoroughly characterized for β_2_-microglobulin. Although the clinical effects are generally described as being due to doxycycline’s ability to disrupt amyloid fibrils, doxycycline could affect many aspects of amyloid formation or patient metabolism. For example, tetracyclines are inhibitors of translation [54] and of matrix metalloproteinases [55], which may both contribute to efficacy. Alternatively, doxycycline could simply be an effective antibiotic in patients with amyloidosis, which is potentially important since treatments directed at antibody-secreting cells can have severe immunosuppressive side effects.

Many small molecules show activity in vitro at micromolar concentrations, but it is not clear that such activity can be extrapolated to patients, where drug absorption and plasma protein binding can significantly reduce the effective concentration of the small molecule, even if toxicity is not limiting. A small, uncontrolled clinical trial suggested improvements in patients [56] with β_2_-microglobulin amyloidosis, even though the plasma concentration of doxycycline in these individuals was only around 2 µM. Doxycycline reduces the rate of aggregation of an amyloidogenic variant of β_2_-microglobulin, as measured by both ThT fluorescence and solubility measurements [34]. However, in these studies, doxycycline inhibited amyloidogenesis of 20 µM β_2_-microglobulin only at concentrations of 50 µM or greater.

The data presented here are not consistent with doxycycline being an inhibitor of LC amyloidogenesis in vitro. Overall, these data support a mechanism of action for doxycycline in AL amyloidosis that does not depend entirely on its ability to prevent amyloidogenesis or disrupt fibrils. Thus, doxycycline may complement both existing cytotoxic agents and novel, non-cytotoxic therapies in patients.

Another molecule that has been studied in detail is EGCG, a bioactive compound found in green tea. Biochemical studies show that EGCG can modulate protein aggregates, in some cases “remodeling” fibrils to form other oligomeric species [35,36]. However, a Phase 2 trial of EGCG in AL amyloidosis failed to show a significant benefit [37] and two other trials (NCT01511263 and NCT02015312) have not reported results.

Small molecules can interfere with biochemical or cell-based assays in multiple ways. Several classes of molecules are regarded as “pan-assay interference” compounds, or PAINS. EGCG is a well-known example: catechols can form reactive orthoquinones upon oxidation [35]. However, pre-oxidation of EGCG did not alter its activity in these assays. Methylene blue, rifamycin SV, and rifampicin all appear to reduce the fluorescence of amyloid-bound ThT. Each of these molecules is colored in solution, a property that should be noted as a potential complication in any optical assay.

The major limitation of this study is that only single examples of LCs were used in each assay. The effects of individual molecules could be unique to specific LC sequences studied. Notably, the study which identified methylene blue as a stabilizer of LCs investigated its binding to the V_L_-domain of a LC derived from the *IGLV2-8* gene [12]. However, the *IGLV6-57* LC gene used in these experiments is over-represented in AL amyloidosis compared to the normal repertoire [41]. The morphologies of the aggregates formed in the presence of small molecules was not investigated further because the physiological relevance of different aggregate structures is not known. Non-amyloid LC aggregates are associated with organ toxicity in light-chain deposition disease and multiple myeloma [57,58]. Another limitation is that only a single concentration of small molecule was used in each experiment. Dose–response studies are critical to differentiating between small molecules. However, on-target activity was only observed for coumarin 1, for which dose–response data have been published [10].

Treatment of AL amyloidosis has benefited enormously from improvements in cytotoxic drugs directed against clonal plasma cells over the last three decades [1]. Further improvements in patient care may be possible by augmenting this strategy with drugs that target different aspects of the disease pathology.

## 4. Materials and Methods

Small molecules were purchased from Sigma (St. Louis, MS, USA) or Thermo Fisher (USA), and 5 mM stocks in water (doxycycline hyclate, EGCG, methylene blue) or ethanol (coumarin 1, diflunisal, rifamycin SV, rifampicin) were prepared and stored at −20 °C until needed. “Oxidized” EGCG was prepared by incubating aliquots of EGCG overnight at 37 °C in water, which were then stored at −20 °C. All experiments were carried out in phosphate-buffered saline (PBS; 1.5 mM KH_2_PO_4_, 8.1 mM Na_2_HPO_4_, 2.7 mM KCl, 138 mM NaCl, pH 7.4) at 37 °C.

Data were analyzed and figures prepared using the “Tidyverse” [59] suite of tools for R [60], within the RStudio environment [61]. Final figures were prepared using Adobe Acrobat.

### 4.1. Light-Chain Preparation

Light chains were cloned into vectors containing T7 promoters and recombinantly expressed in BL21 (DE3) *E. coli*, as previously described [26]. Full-length LCs were expressed as inclusion bodies in the *E. coli* cytosol. Cells were lysed by sonication and insoluble material was washed three times with PBS containing 1% NP-40 detergent, then once with PBS. Inclusion bodies were resuspended in 25 mM Tris-Cl, pH 8 (room temperature pH value), containing 4 M guanidine hydrochloride and 5 mM dithiothreitol for at least 2 h, at 4 °C, before being refolded by dropwise dilution into 25 mM Tris-Cl pH 8 on ice to dilute out the denaturant. V_L_-domains were expressed in the *E. coli* periplasm and extracted by periplasmic shock. Cell pellets were resuspended in 200 mM tris pH 8 containing 0.5 M sucrose and 5 mM EDTA, incubated on ice for 30 min, then shocked by addition of two volumes of deionized water. Cells were removed by centrifugation. Both full-length LCs and V_L_-domains were concentrated by ammonium sulfate fractionation (LCs precipitate in the 25–75% saturation fraction at 4 °C), resuspended in 25 mM Tris-Cl, pH 8, and dialyzed against 25 mM Tris-Cl, pH 8, to remove residual ammonium sulfate. LCs were purified by ion exchange and size exclusion chromatography on Source 15Q and Superdex 75 columns (Cytiva, Marlborough, MA, USA). LCs were eluted from the size exclusion column in PBS, aliquoted, snap-frozen, and stored at −80 °C.

### 4.2. Limited Proteolysis

LCs (5 µM) were incubated at 37 °C for 4 h in the presence of small molecules (50 µM) or appropriate vehicle control, with or without 200 nM proteinase K (Thermo Fisher). Reactions were initiated by adding 2 µL proteinase K or buffer to 18 µL LC solution and incubated in a thermocycler with a heated lid to minimize evaporation. At the end of the incubation time, reactions were quenched with 2 µL of 10 mM phenylmethyl sulfonyl fluoride, which rapidly and irreversibly inactivates the protease. The extent of proteolysis was measured by SDS-PAGE. Loading buffer was added to each sample to a final concentration of 2% SDS, 10% glycerol, and 0.1% 2-mercaptoethanol, and the samples were incubated at 98 °C for 6 min. Samples were run on 16% tris-glycine polyacrylamide gels (Thermo Fisher), stained with GelCode Blue (Thermo Fisher), and visualized using an ImageQuant LAS 4000 gel imaging system (Cytiva). Bands were quantitated with ImageLab software (Bio-Rad, Hercules, CA, USA). Remaining LC was defined as the intensity of the full-length LC band after proteolysis, relative to the un-proteolyzed LC sample, calculated independently for each of three gels.

### 4.3. Amyloidogenesis Kinetics

Thioflavin T (Sigma) was dissolved in ethanol, diluted in PBS to approximately 200 µM, then filtered to remove aggregates. Filtration of ThT often leads to some loss of dye, so ThT concentration was determined by spectroscopy (extinction coefficient of 28,000 at 414 nm) after filtration. Aggregation was carried out in black, clear-bottomed polystyrene microwell plates (item #3631, Corning Life Sciences, Glendale, AZ, USA). JTO-V_L_ was thawed and filtered through a 0.22 µm syringe filter, then added to pre-filtered ThT at final concentrations of 5 µM LC and 5 µM ThT in PBS. The outer wells of the plate were filled with 200 µL PBS to help stabilize the temperature and humidity within the sealed plate and thereby minimize evaporation, leaving 60 wells for LC samples. To each of these wells, 99 µL of LC and ThT solution was added, followed by 1 µL of the appropriate small molecule or vehicle. Small molecule was added to 3 wells per molecule before aggregation, and a further 3 wells were used as controls, where the small molecule was added after aggregation to assess the effect of the molecule on ThT fluorescence. Plates were sealed with film, then covered with a lid that was held in place by adhesive tape. Fluorescence (λex = 440 nm, λem = 480 nm, reading through the bottom of the plate) was measured in a SpectraMax M5 plate-reader (Molecular Devices, San Jose, CA, USA). The plates were incubated quiescently at 37 °C for 10 min to allow the reactions to equilibrate, then an initial reading was taken. Plates were incubated at 37 °C on an orbital plate shaker operating at 1000 rpm. Further readings were taken at regular intervals.

Once aggregation had completed (determined by a plateau in the ThT fluorescence) in all wells without added small molecule, a final fluorescence measurement was taken. The plate was then unsealed and a 1 µL bolus of small molecule solution was added to each of the 3 control wells. Fluorescence was measured again to determine the effect of small molecule on ThT fluorescence. The change in fluorescence upon small-molecule addition was defined as the difference in fluorescence in each control well, corrected for the average change in the wells which had initially contained the same small molecule, which had not otherwise been manipulated between readings, to account for evaporation or other changes during handling.

### 4.4. Filter Trap Assay

Samples were drawn through a cellulose acetate membrane (Sterilitech, Kent, WA, USA) and cellulose nitrate membrane (Bio-Rad, Hercules, CA, USA) using a Bio-Dot vacuum manifold (Bio-Rad). The membranes were pre-wetted with PBS, then 100 µL of sample was drawn through. The protein spots were washed 3 times by drawing 200 µL PBS through the membranes. Membranes were stained with 0.1% (*w*/*v*) amido black in 10% (*v*/*v*) acetic acid, then gently washed with 5% acetic acid and deionized water until the background stain was removed. Aggregated protein adheres weakly to cellulose acetate and can be knocked off by vigorous washing. Spots were imaged using an ImageQuant LAS 4000 gel imaging system (Cytiva) and quantitated with ImageLab software (Bio-Rad), treating each spot as a gel band in order to control the software’s background subtraction feature. The ratios between the background-corrected intensities are reported in Figure 5 and Figure 6 without further normalization.

### 4.5. Fibril Dissociation

A single batch of 6aJL2-V_L_ fibrils was prepared by incubating 5 µM V_L_-domain in PBS at 37 °C, 200 rpm, until visible aggregates formed. Fibrils were vortexed, then split into 500 µL aliquots and centrifuged at 20,000× *g* for 10 min. The supernatant was aspirated to remove any remaining soluble LC, leaving 10 µL in the tubes to avoid disturbing the pelleted fibrils. The fibrils were resuspended in PBS containing 50 µM of small molecule and vortexed. Four samples per molecule were incubated quiescently at 37 °C for seven or fifteen days (2 samples each). The fifteen-day incubation was chosen to replicate similar experiments, where doxycycline was observed to dissociate fibrils [22]. After incubation, samples were centrifuged at 20,000× *g* for 10 min to separate soluble LC from fibrils. The pellets were resuspended in PBS without small molecules and both samples were stored at 4 °C until soluble and insoluble fractions were measured by the filter trap assay, which was carried out as described above. A total of 80 samples was used for the filter trap assay: two replicates each of supernatant and precipitate, at two timepoints, for ten small molecules, including ethanol and water vehicles. This experimental design precluded triplicate experiments, so that all samples could be stained on the same membranes.

## Figures and Tables

**Figure 1 molecules-26-03571-f001:**
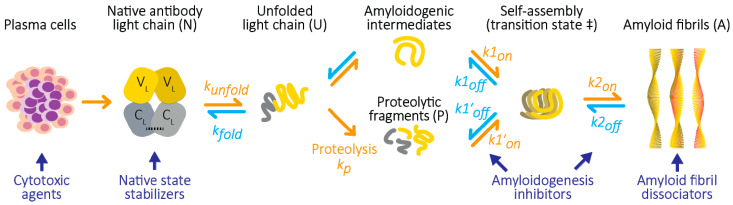
Amyloid formation by antibody light chains. In AL amyloidosis, LCs, shown here as a homodimer, are secreted from a clonal population of plasma cells. Once in circulation, LCs can unfold to amyloid-competent structural states, which may involve proteolysis. Inter-molecular interactions within this population of non-native structures can lead to the formation of polymeric species, including amyloid fibrils. Processes that lead to amyloid formation are shown as orange arrows, while those which antagonize amyloidogenesis are shown as cyan arrows. Reversible processes are shown as equilibria. Reaction rates for these processes can in some cases be calculated. The folding and unfolding rate constants of the native LC are defined as *k_fold_* and *k_unfold_*, respectively. The self-assembly process is divided into initiation and elongation steps, with rate constants *k1_on/off_* and *k2_on/off_*, respectively. Proteolysis is irreversible with a rate constant of *k_p_*, and leads to self-assembly of fragments with rate constants *k1′_on/off_*. LCs can form multiple fibril structures. Therapies for AL amyloidosis (blue text and arrows) could potentially target any of these steps, which implies that combining therapies with different mechanisms of action could benefit patients. Existing treatments aim to kill the clonal cells that secrete the LCs. Alternative treatments could suppress formation of non-native amyloidogenic LC species (stabilizers), inhibit self-association (amyloidogenesis inhibitors), or promote the dissolution or removal of existing amyloid fibrils (amyloid dissociators).

**Figure 2 molecules-26-03571-f002:**
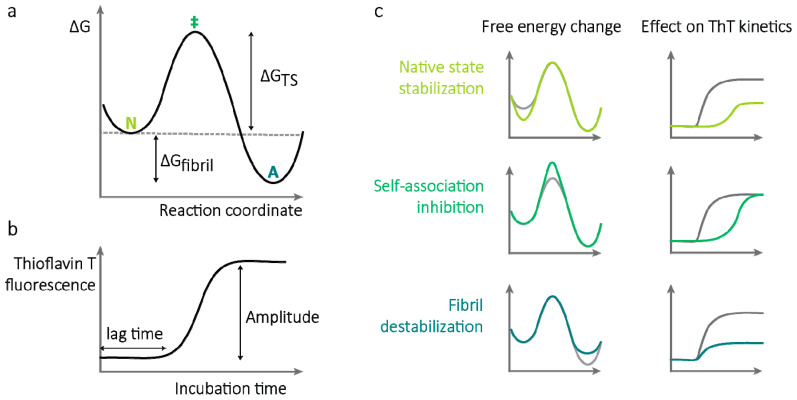
Mechanisms by which small molecules could inhibit amyloidogenesis. (**a**) Simple model of aggregation, where native (N) and amyloid (A) states are separated by a single transition state (‡), which represents the complex process of self-association of amyloid-competent LCs. Two free energy differences define the behavior of this system: ΔG_fibril_ is the free energy difference between the native and fibril states, which defines the equilibrium concentrations of these two states, and ΔG_TS_ is the free energy barrier to aggregation (or disaggregation), which defines the rate at which the system reaches equilibrium. Note that this model conflates initiation and extension of amyloid fibrils. ΔG_fibril_ defines the critical concentration, or solubility limit, of a protein. (**b**) Expected results from an aggregation kinetics measurement using thioflavin T (ThT) fluorescence. The steep rise in fluorescence is due to the increased rate of fibril formation in the presence of existing fibrils. If the simple model depicted in panel (**a**) is correct, the ThT fluorescence amplitude reports on ΔG_fibril_ and the lag time reports on ΔG_TS_. (**c**) The effect of small molecules on aggregation kinetics can indicate the mechanisms by which the molecules affect the aggregation reaction. Green lines show the predicted effects of each mechanism. Stabilizing the native state (top) should increase the lag time and decrease the fluorescence amplitude. Destabilizing the transition state (middle) should increase the lag time but not affect the endpoint fluorescence. Destabilizing the fibrils (bottom) should decrease the endpoint fluorescence amplitude but not affect the lag time.

**Figure 3 molecules-26-03571-f003:**
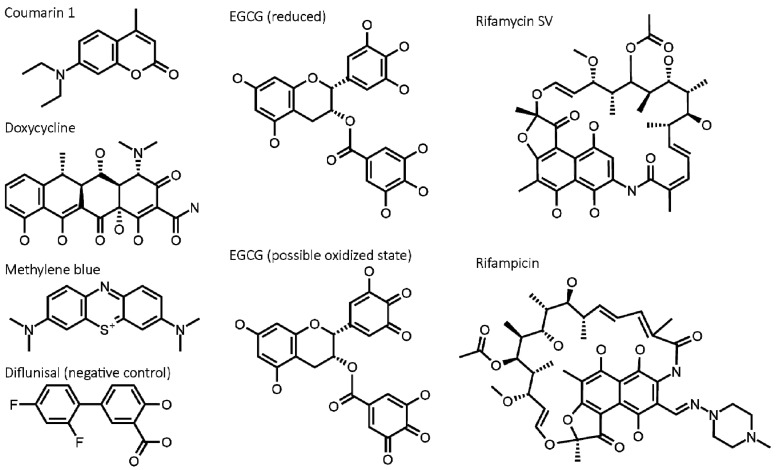
Structures of small molecules used in this study. The natural product EGCG can undergo oxidation to form reactive orthoquinone species such as that shown. To generate these species, EGCG solutions were incubated for 24 h at 37 °C in water. Diflunisal, a stabilizer of transthyretin, is not expected to interact with LCs and is included as a negative control.

**Figure 4 molecules-26-03571-f004:**
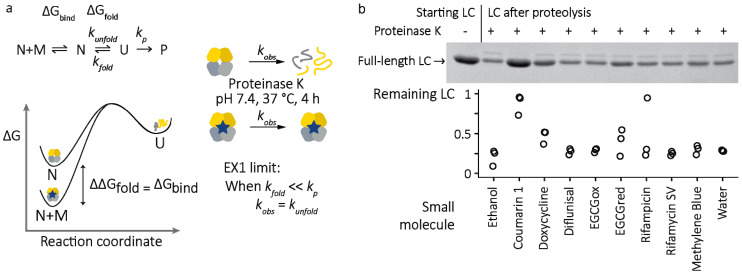
Stabilization of LCs measured by proteolysis. (**a**) Experimental outline. Transient unfolding of native LCs (N) to protease-sensitive conformations (U) limits the rate at which the LCs are cleaved by proteinase K. Binding of small-molecule ligands (M) stabilizes LCs against unfolding and therefore reduces the rate of proteolysis. (**b**) Residual full-length LC after incubation with proteinase K and small molecules was measured by SDS-PAGE (*n* = 3 independent reactions per molecule). Top, example gel showing the starting material (**left** lane) and remaining LC after incubation (**right** lanes). Bottom, quantitation of gel bands, relative to the un-proteolyzed LC.

**Figure 5 molecules-26-03571-f005:**
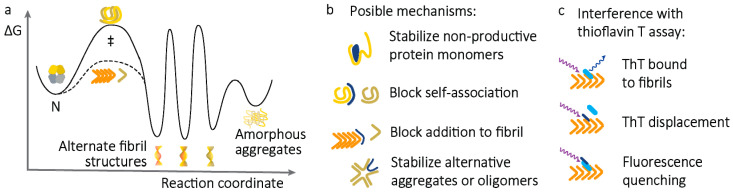
Schematic depiction of amyloid formation by non-native LCs. (**a**) The endpoint proportion of aggregated LCs is defined by the critical concentration, but the rate at which this reaction proceeds is determined by the self-association of amyloid-competent, non-native LCs, associated with the transition state ‡. The energy barrier associated with extension of fibrils (dashed line) is lower than that for nucleation of new fibrils. Multiple fibril or other aggregate structures may be accessible to the LCs, shown as distinct energy minima. However, the energy barriers to interconversion between these states are likely to be large, requiring dissociation, unfolding, and reaggregation of the subunit molecules. (**b**) Small molecules could inhibit amyloid formation (or enhance amyloid dissociation) via multiple mechanisms, depicted schematically. (**c**) Amyloid formation can be readily measured by binding of the fluorogenic dye thioflavin T (shown as a cyan bar), but small molecules (black bars) may interfere with this assay. Excitation and emission photons are shown as wavy arrows.

**Figure 6 molecules-26-03571-f006:**
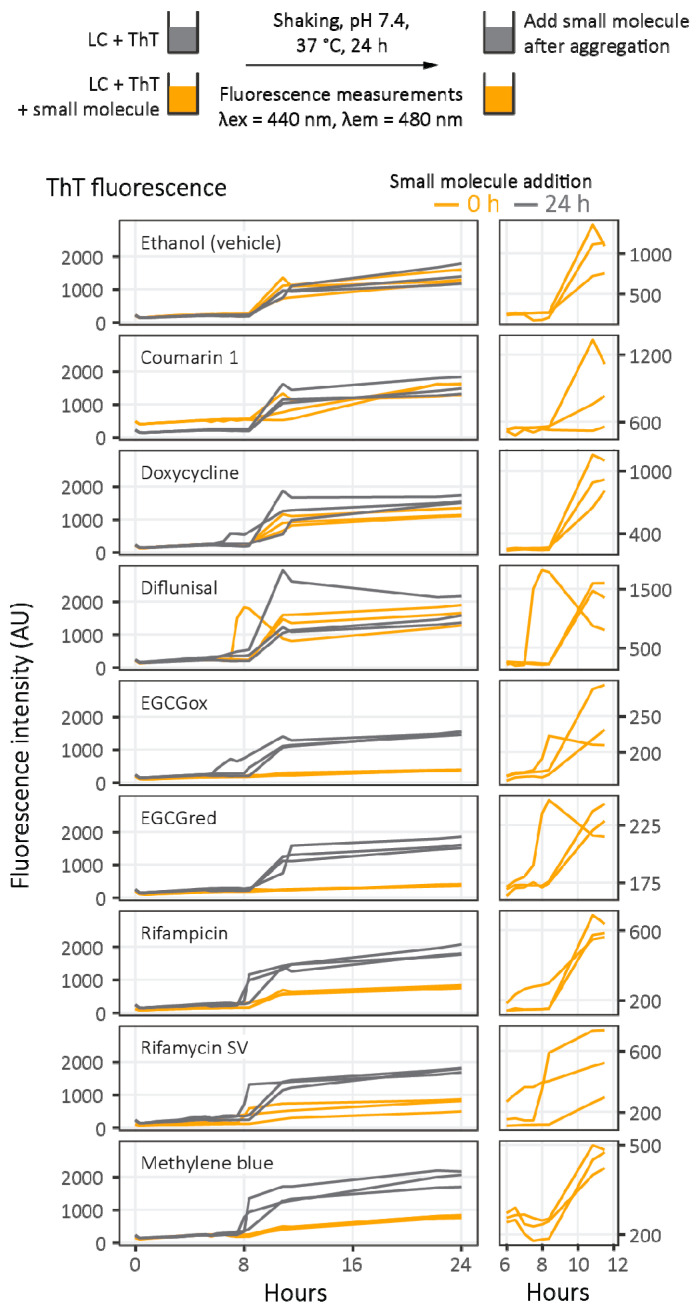
Aggregation of LC V_L_-domains measured by thioflavin T fluorescence. Experimental design is shown at the top. After 24 h of aggregation monitored by ThT fluorescence, the effect of small-molecule addition to pre-formed fibrils on ThT fluorescence was measured (Figure 7, below), and the amount of soluble and aggregated LC is quantitated by the filter trap assay (Figure 8, below). Orange traces represent fluorescence intensity changes (*n* = 3 wells) in the presence of small molecules, and grey traces represent control wells where small molecules were added after 24 h (*n* = 3 wells). Panels on the right show the detail of the time period where aggregation occurs.

**Figure 7 molecules-26-03571-f007:**
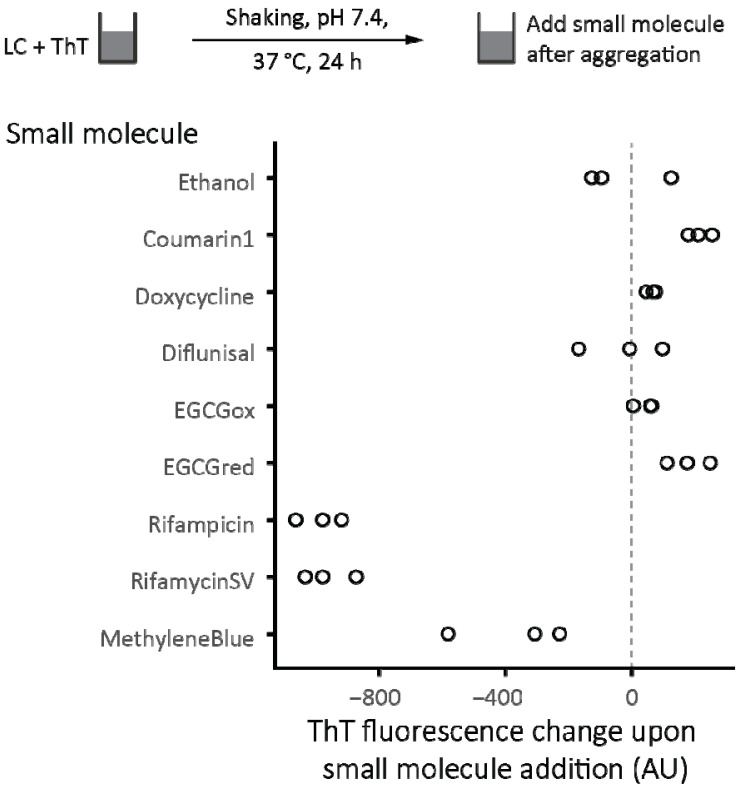
The effect of addition of small molecules to pre-formed, ThT-bound fibrils. Fibril formation was monitored by ThT fluorescence as described in Figure 6. The decrease in ThT fluorescence shown for rifampicin, rifamycin SV, and methylene blue is consistent with fluorescence quenching or displacement of ThT from fibrils.

**Figure 8 molecules-26-03571-f008:**
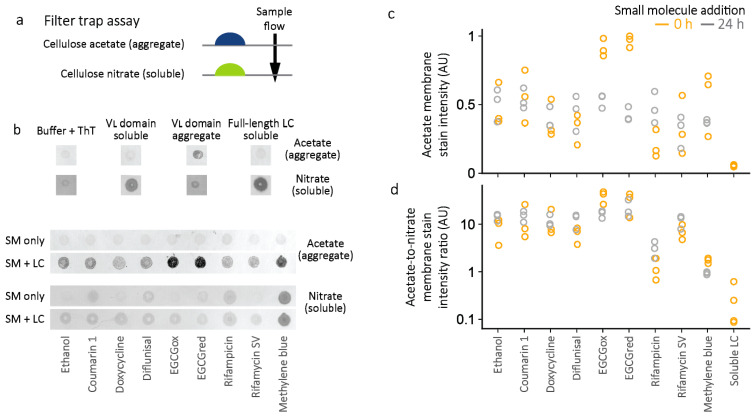
Quantitation of soluble and aggregated LCs by the filter trap assay. Fibril formation was monitored by ThT fluorescence as described in Figure 6. (**a**) Schematic diagram of the filter trap assay. Soluble proteins pass through the cellulose acetate membrane and bind to the cellulose nitrate membrane. (**b**) Examples of stained membranes. Control samples (top) show the staining of soluble or aggregated LCs on cellulose acetate or cellulose nitrate membranes. Two types of soluble LCs were used as negative controls for aggregation: JTO-V_L_ that had not been incubated at 37 °C and full-length JTO that did not aggregate over the course of the experiment. Note that several small molecules, especially methylene blue, stain the membranes in the absence of protein, which affects their apparent quantitation. (**c**) Aggregate load is estimated by quantitation of the cellulose acetate membrane staining, relative to the most intense sample. Orange points represent wells where small molecules were present throughout the aggregation process, and grey points represent wells where the small molecules were added after aggregation had occurred. (**d**) The ratios of aggregated to soluble LCs are shown for each individual sample. Symbols are colored as for panel (**c**).

**Figure 9 molecules-26-03571-f009:**
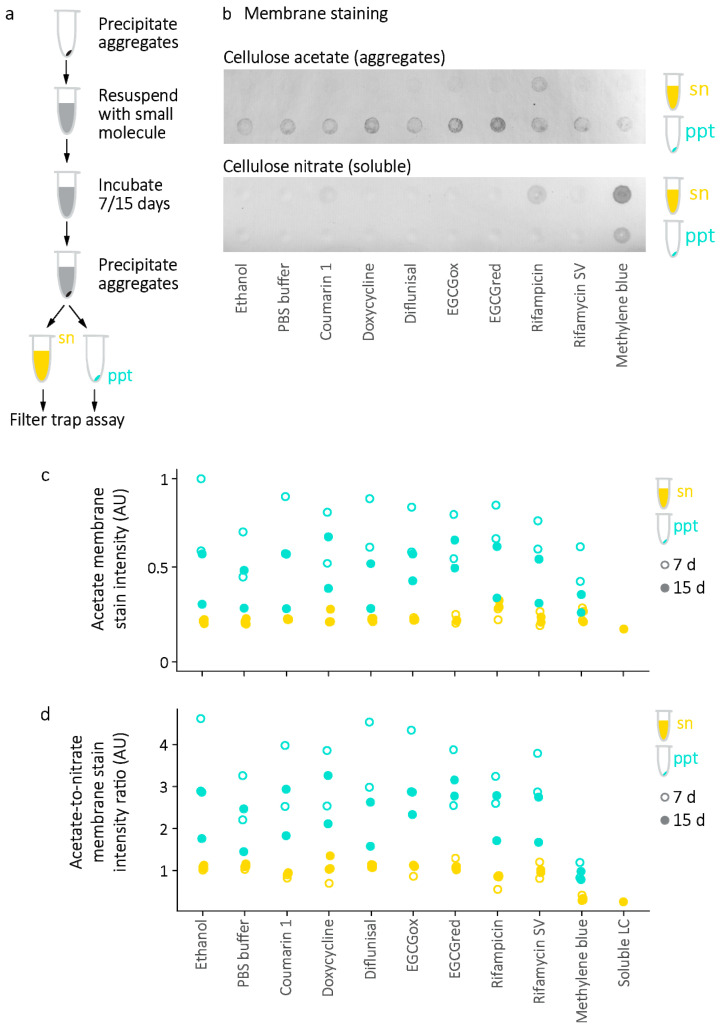
Dissociation of pre-formed amyloid fibrils measured by the filter trap assay. (**a**) Experimental outline. Pre-formed fibrils (*n* = 2 samples per condition) were incubated with small molecules for 7 or 15 days, after which bulk aggregated LCs were precipitated by centrifugation. The soluble and insoluble LC in the supernatant (sn, orange) and precipitate (ppt, blue) were measured by the filter trap assay. (**b**) Example membranes showing aggregated (cellulose acetate membrane stain) and soluble (cellulose nitrate membrane stain) LC in the supernatant and precipitate of one set of samples incubated for 7 days. (**c**) Total aggregate levels in each sample, relative to the most-stained membrane region. (**d**) Ratio of aggregated to soluble LC after 7 days (hollow symbols) or 15 days (solid symbols). The reduced apparent aggregation observed in the presence of methylene blue appears to be due to staining of the cellulose nitrate membrane by the small molecule. Symbols are colored as for panel (**c**).

**Table 1 molecules-26-03571-t001:** Potential strategies to inhibit LC amyloidogenesis. Note that the mechanisms of action of doxycycline and epigallocatechin gallate (EGCG) are not known.

Strategy	Examples	Comments
Cytotoxic agents	Melphalan and stem cell transplant Proteasome inhibitors Anti-CD38 antibodies	In routine clinical use. Suppress LC secretion by killing the producer cells. Significant side-effects.
Native state stabilizers	Coumarin 1 Methylene Blue	In vitro studies, no clinical data for AL amyloidosis.
Amyloidogenesis inhibitors	Doxycycline? EGCG?	Proposed mechanism. Clinical trials in progress.
Amyloid fibril disruptors	Doxycycline? EGCG?	Proposed mechanism. Clinical trials in progress.
	Anti-amyloid antibodies	Clinical trials in progress. Limited efficacy observed to date, possibly due to diverse fibril structures.

**Table 2 molecules-26-03571-t002:** Summary of results. Values represent mean ± standard deviation, *n* = 3 replicates. EGCGox and EGCGred refer to pre-oxidized and freshly thawed EGCG, respectively.

Molecule	Stabilization of Full-Length WIL LC (Remaining LC)	JTO-V_L_ Endpoint ThT Fluorescence (Relative to Vehicle)	Inhibition of JTO-V_L_ Aggregation (Filter Trap)	Dissolution of 6aJL2-V_L_ Fibrils after 15 Days (Filter Trap)
Coumarin 1	0.88 ± 0.13	0.97 ± 0.12	No	No
Doxycycline	0.28 ± 0.037	0.94 ± 0.18	No	No
Diflunisal	0.47 ± 0.085	0.75 ± 0.08	No	No
EGCGox	0.29 ± 0.025	0.25 ± 0.01	No	No
EGCGred	0.4 ± 0.17	0.23 ± 0.02	No	No
Rifampicin	0.49 ± 0.4	0.42 ± 0.03	No	No
Rifamycin SV	0.25 ± 0.025	0.41 ± 0.11	No	No
Methylene blue	0.3 ± 0.057	0.41 ± 0.02	No	No

## Data Availability

All data collected are presented in the main figures.

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
