# Peer review of "Barriers to Small Molecule Drug Discovery for Systemic Amyloidosis"

_molecules, 2021, doi:10.3390/molecules26123571_

Round 1
Reviewer 1 Report
Manuscript “Barriers to small molecule drug discovery for systemic amyloidosis” is dedicated to establishing the effects of 8 selected model compounds on fibrillization of antibody light chain (LC), which is a model of amyloid light chain amyloidosis. Effects of compounds on stabilization of native conformation, inhibition of fibrillization, and dissociation of preformed fibrils are studied at one concentration of effectors. The author shows that non of the selected compounds can inhibit fibrillization or dissolve LC fibrils, which is interpreted as difficulty in drug discovery for systemic amyloidosis.
We feel that there are serious shortcomings in the study design and performing of experiments, which need to be improved. First – in selecting of compounds it is considered that fibrillization inhibitors are universal. Amyloid formation is specific and therefore no universal fibrillization inhibitors exist. Second - screening of 8 compounds at one concentration towards one object is definitely not sufficient nor conclusive to decide about the feasibility of an anti-amyloid drug design strategy. Third - figures are at first glance attractive but after analysis, they seem to be oversimplified especially in thermodynamic interpretations. Fourth - it's a strong point of study that different aspects of fibrillization inhibition are considered and studied, however, all techniques used are semiquantitative and linked with substantial experimental error.
It would be informative to start with cuvette experiments of fibrillization to get reproducible sigmoidal fibrillization curves and effects of selected compounds on lag phase and elongation phase of fibrillization. It is correctly stated that some compounds can decrease ThT fluorescence and it should also be studied with preformed fibrils. Fibril dissolution experiments are unnecessary as according to the general view the protein fibrils are thermodynamically so stable that small molecules cannot dissolve them. Some TEM experiments would also be needed to confirm the formation of fibrils in the case of LC-s studied. More compounds should be studied before drawing general conclusions about the anti-amyloid drug design strategy.
Author Response
The comments from Reviewer 1 are shown in black
The responses are shown indented, and in red.
Manuscript “Barriers to small molecule drug discovery for systemic amyloidosis” is dedicated to establishing the effects of 8 selected model compounds on fibrillization of antibody light chain (LC), which is a model of amyloid light chain amyloidosis. Effects of compounds on stabilization of native conformation, inhibition of fibrillization, and dissociation of preformed fibrils are studied at one concentration of effectors. The author shows that non of the selected compounds can inhibit fibrillization or dissolve LC fibrils, which is interpreted as difficulty in drug discovery for systemic amyloidosis.
I thank the reviewer for their helpful and insightful comments. I clearly had not explained the aims and scope of the manuscript well enough, because although I agree with most of the reviewer’s points, our conclusions are quite different. I have extensively rewritten the manuscript to try to rectify this.
We feel that there are serious shortcomings in the study design and performing of experiments, which need to be improved. First – in selecting of compounds it is considered that fibrillization inhibitors are universal. Amyloid formation is specific and therefore no universal fibrillization inhibitors exist.
I did not mean to suggest that “universal” inhibitors exist – this was not stated in the original manuscript and I apologize that this was the impression that was given. This issue is now discussed in more detail in lines 115-125 in the revised manuscript. Although I agree with the reviewer that there are unlikely to be universal inhibitors, the published observations on doxycycline and EGCG appear to have been interpreted in that way. There is reasonable evidence that doxycycline inhibits amyloid formation by beta-2-microglobulin at high concentrations in vitro, and the original report of anti-amyloid tetracycline activity studied ATTR fibrils. The evidence for activity against light chain fibrils is weaker, yet there is some suggestion of clinical benefit in AL amyloidosis. This manuscript asks whether doxycycline and other molecules suppress aggregation in our experimental systems, and finds that they do not.
Second - screening of 8 compounds at one concentration towards one object is definitely not sufficient nor conclusive to decide about the feasibility of an anti-amyloid drug design strategy.
I agree, but this was not the purpose here. I aimed to look for activity of molecules that had been described as active against Ig-fold proteins in the literature. Strong evidence that doxycycline inhibits light chain aggregation or remodels fibrils would have led to a different study, but I think that these negative results should be reported.
Third - figures are at first glance attractive but after analysis, they seem to be oversimplified especially in thermodynamic interpretations.
The thermodynamics are deliberately simplified, since the experiments here are not able to measure free energies with the precision necessary to draw accurate diagrams.
Fourth - it's a strong point of study that different aspects of fibrillization inhibition are considered and studied, however, all techniques used are semiquantitative and linked with substantial experimental error.
Precise, quantitative measures of amyloid formation are difficult to scale sufficiently to investigate multiple small molecules, especially given the noise inherent in these systems. I have attempted to quantify the data as much as possible, and have highlighted This approach is also to ask whether these molecules have a sufficiently large effect to be detected in high-throughput screens.
It would be informative to start with cuvette experiments of fibrillization to get reproducible sigmoidal fibrillization curves and effects of selected compounds on lag phase and elongation phase of fibrillization.
Cuvette data have been published for JTO-VL, originally by Wall et al. in 1999. We have observed significant day-to-day variation with these experiments, so prefer to use plates for direct comparisons between conditions, even though the data are less precise.
It is correctly stated that some compounds can decrease ThT fluorescence and it should also be studied with preformed fibrils.
This is the experiment now shown in Figure 7 (previously Figure 5c). I am not sure what a more detailed investigation of the loss of ThT signal would add here.
Fibril dissolution experiments are unnecessary as according to the general view the protein fibrils are thermodynamically so stable that small molecules cannot dissolve them.
This is precisely the hypothesis that the paper sets out to test. I agree with the reviewer that dissolution of fibrils is unlikely, and indeed this is what was observed, but doxycycline and EGCG have repeatedly been reported as doing exactly that. By focusing on aggregate mass balance, rather than thioflavin T fluorescence, I aimed to determine whether this dissolution really occurs. As the manuscript states, we do not know which aggregate species are toxic in vivo, and whether the "remodeling" observed in vitro is possible in patients.
Some TEM experiments would also be needed to confirm the formation of fibrils in the case of LC-s studied.
Electron micrographs of JTO-VL fibrils have been published previously. Because EM is not strictly quantitative, it is difficult to show the absence of fibrils and exclude the possibility that any treatment does not alter adhesion to the grid.
More compounds should be studied before drawing general conclusions about the anti-amyloid drug design strategy.
I agree and have tried to avoid making general statements. Again, this manuscript aims to ask specifically about small molecules which have been described in the literature as inhibiting Ig-domain amyloidogenesis and I hope that I have made this more clear. I do note that we have successfully identified light chain stabilizer molecules using a high-throughput screening approach.
Reviewer 2 Report
Comments to authors
The manuscript entitled “Barriers to small molecule drug discovery for systemic amyloidosis” by Gareth, described the interesting aspects of current development of small-molecules and problems to treat amyloidosis with a particular interest on AL. In general, the author probed the effect of small molecules on (i) AL proteolysis, (ii) inhibition of amyloid aggregation and (iii) fiber dissolution. By using a number of small molecules that are tested in the past for several other diseases, the author reported coumarin 1 protects LC against proteolysis; however, others have no effect. Similarly, using a routinely used amyloid dye used to monitor amyloid aggregation kinetics, the author showed none of these small molecules potentially inhibit LC aggregation or dissolve preformed AL fibers.
This referee found that this article well written and important topics in the field and would be interesting for amyloid researchers. Therefore, this referee recommends publication of this article after suggested changes.
- AL amyloidosis should be defined in the abstract.
- Figure 1 captions: Kon/off and fold and unfold are italicized, whereas in the figure non-italic symbols are used. A uniform presentation is recommended. Drugs, inhibitors and amyloid dissociators are presented with an identical symbol (T). Small-molecule inhibitors, drugs and dissociators differ in their chemical and physical properties. For instance, nanoparticles have been developed as potential amyloid inhibitors and dissociators but are not considered as drugs. This author recommends revision for figure 1.
- Figure-1, the free energy graphs are not well defined. Define N, U, +, and A.
- Fibril structures shown in figure-1 and referred in caption are hard to follow. It would benefit readers by expanding the number of labels for individual sub-figures in Figure-1.
- The paragraph (Lines 106-123) summarizing different approaches to control AL is important in perspective of this article. This reviewer recommends a figure outlining different approaches, current success and limitations in the development of small molecules to control AL.
- Figure-5: The representation of graphs in figure-5 needs revision. Define Y-axis units for individual graphs.
- Several sentences in results and discussion section are not supported with references. This reviewer suggests to include citation in appropriate sections. For example, “A small molecule which preferentially reacts ……molecular ratchet” mechanism”. “Doxycycline has been reported to dissociate amyloid fibrils over a 15-day time period.”
- Lastly, the author finds the current manuscript hard to follow due to the data representation. A systematic dislagging of the presented results would benefit the readers to follow the presented study.
Author Response
Comments from Reviewer 2 are in black
Responses are shown indented, and in red
The manuscript entitled “Barriers to small molecule drug discovery for systemic amyloidosis” by Gareth, described the interesting aspects of current development of small-molecules and problems to treat amyloidosis with a particular interest on AL. In general, the author probed the effect of small molecules on (i) AL proteolysis, (ii) inhibition of amyloid aggregation and (iii) fiber dissolution. By using a number of small molecules that are tested in the past for several other diseases, the author reported coumarin 1 protects LC against proteolysis; however, others have no effect. Similarly, using a routinely used amyloid dye used to monitor amyloid aggregation kinetics, the author showed none of these small molecules potentially inhibit LC aggregation or dissolve preformed AL fibers.
This referee found that this article well written and important topics in the field and would be interesting for amyloid researchers. Therefore, this referee recommends publication of this article after suggested changes.
I thank the reviewer for their comments and suggestions. The manuscript has been extensively rewritten, to address the points raised by both reviewers.
AL amyloidosis should be defined in the abstract.
Done
Figure 1 captions: Kon/off and fold and unfold are italicized, whereas in the figure non-italic symbols are used. A uniform presentation is recommended. Drugs, inhibitors and amyloid dissociators are presented with an identical symbol (T). Small-molecule inhibitors, drugs and dissociators differ in their chemical and physical properties. For instance, nanoparticles have been developed as potential amyloid inhibitors and dissociators but are not considered as drugs. This author recommends revision for figure 1.
Figure 1 has been simplified and standardized. I use “drugs” as a generic term for medicines, and now refer to “cytotoxic agents” in figure 1. (I think that regulatory authorities would consider anti-amyloid nanoparticles as drugs, rather than devices.) Rather than the “inhibitor” symbol, the targets of the therapies are now shown with arrows.
Figure-1, the free energy graphs are not well defined. Define N, U, +, and A.
Now defined in Figure 2, and also in Figures 4 and 5.
Fibril structures shown in figure-1 and referred in caption are hard to follow. It would benefit readers by expanding the number of labels for individual sub-figures in Figure-1.
Figure 1 has been simplified to rectify these issues.
The paragraph (Lines 106-123) summarizing different approaches to control AL is important in perspective of this article. This reviewer recommends a figure outlining different approaches, current success and limitations in the development of small molecules to control AL.
This information is now included in Table 1. I tried to include it in Figure 1 but could not find a pleasing way to incorporate the extra text.
Figure-5: The representation of graphs in figure-5 needs revision. Define Y-axis units for individual graphs.
The y-axes for the graphs (now in Figure 8) are have been clarified.
Several sentences in results and discussion section are not supported with references. This reviewer suggests to include citation in appropriate sections. For example, “A small molecule which preferentially reacts ……molecular ratchet” mechanism”. “Doxycycline has been reported to dissociate amyloid fibrils over a 15-day time period.”
I removed the discussion of covalent compounds and molecular ratchet mechanisms, cited the relevant reference for doxycycline, and have added additional references through the revised manuscript.
Lastly, the author finds the current manuscript hard to follow due to the data representation. A systematic dislagging of the presented results would benefit the readers to follow the presented study.
The manuscript has been extensively rewritten and the figures positioned by hand, rather than the automatic conversion to Molecules format. I hope that these revisions help.
Round 2
Reviewer 1 Report
The manuscript has been substantially changed and improved.